# Thermal Water Prospection with UAV, Low-Cost Sensors and GIS. Application to the Case of La Hermida

**DOI:** 10.3390/s22186756

**Published:** 2022-09-07

**Authors:** Javier Sedano-Cibrián, Rubén Pérez-Álvarez, Julio Manuel de Luis-Ruiz, Raúl Pereda-García, Benito Ramiro Salas-Menocal

**Affiliations:** Grupo de Investigación de Ingeniería Cartográfica y Explotación de Minas, Escuela Politécnica de Ingeniería de Minas y Energía, Universidad de Cantabria, Boulevard Ronda Rufino Peón, 254, Tanos, 39300 Torrelavega, Spain

**Keywords:** geothermal, photogrammetry, drone, infrared, hydrology, thermal mapping, reclassification

## Abstract

The geothermal resource is one of the great sources of energy on the planet. The conventional prospecting of this type of energy is a slow process that requires a great amount of time and significant investments. Nowadays, geophysical techniques have experienced an important evolution due to the irruption of UAVs, which combined with infrared sensors can provide great contributions in this field. The novelty of this technology involves the lack of tested methodologies for their implementation in this type of activities. The research developed is focused on the proposal of a methodology for the exploration of hydrothermal resources in an easy, economic, and rapid way. The combination of photogrammetry techniques with visual and thermal images taken with UAVs allows the generation of temperature maps or thermal orthomosaics, which analyzed with GIS tools permit the quasi-automatic identification of zones of potential geothermal interest along rivers or lakes. The proposed methodology has been applied to a case study in La Hermida (Cantabria, Spain), where it has allowed the identification of an effluent with temperatures close to 40 °C, according to the verification measurements performed on the geothermal interest area. These results allow validation of the potential of the method, which is strongly influenced by the particular characteristics of the study area.

## 1. Introduction

Water is fundamental for the survival and development of society. The global demand for water grows annually around 1% [1]. The activities of management and monitoring of the resource are therefore fundamental today. Water quality can be assessed on the basis of a number of parameters such as temperature, oxygen dissolution, etc. [2,3]. Their alteration affects habitats and ecosystems. However, in certain specific cases, unusual values of these parameters can have positive impacts on society.

Water temperature is a parameter of great interest and can be affected by various factors, whether natural or human [4]. One of the natural causes of temperature alteration is derived from the geothermal resource due to the heat of the earth’s interior [5]. This renewable energy has a potential of 2077 MW [6,7]. The geothermal resource is an alternative with great potential to fossil fuels as it is a renewable, clean, and stable source that is independent of weather conditions [8] and is also able to cover the power and heat demands [9]. Despite its multiple benefits, it currently accounts for only about 1% of the renewable energy mix. However, not all the geothermal effluent areas have the same potential to be exploited in technically and economically profitable ways [10]. According to the temperature and enthalpy of the effluent, the geothermal resource can be classified as low (T < 90 °C), medium (90 °C < T < 150 °C), or high temperature (T < 150 °C) [11]. When the geothermal resource in an area has low temperatures, it is usually used directly, in applications where the hot water spring is not intended to produce electricity [10], but to heat processes and spaces in industries or agriculture, for recreational use, spas, leisure, or geothermal tourism [6]. In addition to the importance of water resources, this implies the need for strategies that allow a limited and optimized use of water, especially in the case of geothermal plants [9].

The conventional prospecting or exploration of geothermal resources is a complicated and costly task, involving borehole drilling, geophysical surveys, etc. [12]. However, it is possible to identify geothermal reservoirs from surface temperature measurements, in order to determine potential locations prior to investing in more specialized and expensive activities. Throughout history, various techniques have been used to measure water temperature. The most conventional techniques are based on sampling or spot measurements. However, since the end of the 20th century, the potential of the application of remote sensing and thermal infrared (TIR) techniques has been demonstrated [13], due to the advantages of being non-invasive thermographic imaging techniques, their spatial and temporal flexibility, or the interpretation of the results [14]. This type of technology allows the determination of the surface temperature of water bodies. In the beginning, thermal images from satellites were used for applications such as hydrological modeling, fire detection, or the exploration of mineral resources [15], to gradually give way to newer technologies, such as manned aerial aircrafts [16], or more recently, the unmanned aerial vehicles (UAVs) [17]. These platforms have been a revolution in many engineering fields, demonstrating their potential for a broad range of applications. UAVs make it possible to achieve accuracies of the order of a centimeter, compared to the metric accuracies of satellite imagery [18], while providing greater flexibility, temporariness, and lower prices than manned aircrafts [19,20]. In addition to all this, their already demonstrated potential of use in the field of hydrogeology, in applications such as wetland restoration [21], groundwater discharge survey [22], identification of water bodies extent [23] or geothermal springs mapping [24], should be taken into account.

The first UAVs used only sensors in the visible or RGB spectrum. In recent years, technological advances have allowed a wide variety of sensors to be mounted on this type of equipment, such as multispectral, hyperspectral, or thermal ones. The first investigations developed with thermal sensors and UAVs used handheld TIR cameras which were modified to be integrated into the platform [25]. This was an inefficient alternative due to their heavy weight. In recent years, the miniaturization of thermal sensors has led to the development of a wide variety of equipment [26]. Despite the loss of accuracy and sensitivity associated with the miniaturization process [27], the use of low-cost equipment is increasing in many areas.

Photogrammetry is a technique that allows the reconstruction of 3D models from images [28,29]. The use of TIR images with UAVs has been tested, allowing the generation of three-dimensional representations, such as DTMs, and two-dimensional representations, such as orthomosaics [30]. The use of thermal imaging in the field of photogrammetry is relatively recent, but authors have already called this technique “thermal photogrammetry”. It has been successfully used for water bodies mapping, specifically for the generation of large temperature maps, which allow the identification and monitoring of geothermal resources [6]. The combination of UAV and photogrammetry can produce large amounts of geospatial information in the form of maps and other products for analysis and management, with high spatial and temporal resolution. These activities are possible thanks to the capabilities of the geographic information systems (GIS) [31]. They allow the processing of the generated products, facilitating the exploration and monitoring of the resource. These GIS tools have the potential to analyze georeferenced information through multi-criteria analysis [32] for the identification of potential locations of the geothermal resources.

The combination of these types of techniques (photogrammetry, infrared sensors, UAV, and GIS) has been used in other investigations with positive results in their fields. Several previous works can be presented as examples of their use for the prospection of mineral resources based on the surface temperature of the lithologies existing in the outcrop [33], the identification and characterization of cold-water patches (CWPs) [34], the analysis of the stability of cliffs using UAV and thermal sensors [35], the detection of submarine groundwater discharge (SDG) on the coast [36], the restoration of wetlands [21], the mapping of the surface temperature of a glacier [37], the identification and mapping of a geothermal feature [6], the understanding of transpiration processes and plant stress [38], or the mapping and modeling of an active volcano by means of UAV [39]. These investigations and many more that can be found in the current literature employ similar methodologies, combining in many cases photogrammetry with visible and thermal imaging for the identification or analysis of thermal anomalies on the land surface or water bodies. There are many similarities between the workflows for data processing proposed by the various authors. However, none of them is focused on the identification of hydrothermal resources, where the implementation of UAV thermography requires a simple work methodology that allows the standardization of the prospective activities for this type of resources.

This research proposes the validation of a quasi-automatic methodology for the identification and management of hydrothermal resources in rivers. In addition, it will be applied to a specific case, the recognized hydrothermal exploitation of La Hermida (Cantabria, Spain), on the basis of thermal orthomosaics generated with blocks of thermal images taken with UAV, and their subsequent treatment with GIS tools.

## 2. Methodology

A methodology for the identification of potential zones with hydrothermal resources in rivers is proposed. The methodology is based on the interpretation and analysis using GIS tools of thermal orthomosaics generated by means of the photogrammetric reconstruction of RGB and TIR images obtained by dual sensors integrated in UAVs.

### 2.1. Thermal Imaging with UAVs and Data Capture

The thermal infrared (TIR) and RGB images are the basis for the generation of the thermal orthomosaics that allow the identification of possible geothermal resources in rivers. The acquisition of the images is the first phase of this methodology, and the choice of both the UAV platform and the TIR sensor is its first proposal. Following the premises of this research, the use of a low-cost dual sensor is recommended. This type of device features a non-metric RGB sensor and an infrared one [40], which allows both types of images to be captured from the same location.

Once the equipment for image acquisition has been selected, the flight plan must be designed. In other words, the set of guidelines that determine the way and manner in which the UAV moves over the study area and the sensor captures the images must be determined. Each flight plan is a particular case that must be individually designed to suit the characteristics of the object of study and the required needs [41]. In the particular case of capturing TIR images with this type of devices, it must be considered that TIR sensors present more limiting features than RGB sensors, mainly due to the lower resolution, contrast, and size of the sensor [42]. Therefore, the recommendations for photogrammetric flights vary slightly from those traditionally applied in photogrammetry. Conventionally, the flight plan is designed automatically in one of the many specific programs available [43], only by entering the study area, sensor characteristics, ground sample distance (GSD), frontlap, sidelap, and flight speed.

The GSD is a parameter directly dependent on the flight altitude and the characteristics of the sensor. Due to the TIR sensors features, it is advisable to use lower flight altitudes, in order to obtain a higher level of detail and reach sufficient GSD values for the object of interest. Similarly, the frontlap and sidelap values should be increased to at least 80 and 60%, respectively [44], to eliminate occlusion zones. Ultimately, the flight speed is fundamental to determine the total flight time, which is conditioned to some extent by the autonomy of the UAV [45]. In this case, it is proposed to use reduced flight speeds of 3–4 m/s or even lower [46] to avoid the presence of blurred frames, thus increasing the quality of the TIR images obtained. In addition, the sensor tilt angle should be defined. It is recommended to take nadir images, because oblique angles reduce the emissivity of the object and provide errors in the temperature measurement [47]. In addition, TIR sensors allow setting the emissivity, which depends on various factors. However, for the determination of the temperature of the water surface, a value between 0.96 and 0.99 is generally established [48].

The flight schedule (date and time) is another fundamental parameter, which depends on the weather conditions. It is interesting to arrange the flight on days with medium-low temperatures in order to ensure a temperature difference between the geothermal resource and the surrounding areas. The river current must also be taken into account, in such a way that the periods of maximum flow are avoided, since they may involve a rapid mixing of the emerging hot water and the cold water of the current itself, so that the temperature of the upper layer is not greatly affected, thus preventing the identification of the temperature of the effluent, as the TIR sensors measure only the surface temperature of the water body [49].

Ultimately, it is recommended to control the flight with ground control points (GCP) for a correct georeferencing of the products obtained, and to facilitate the image alignment during the corresponding photogrammetric processing, especially when working with TIR images, since they are single-band images with lower resolution [50]. The distribution of the GCPs in the study area is one of the parameters with the greatest influence on the accuracy obtained [51], so a regular and uniform distribution throughout the area to be represented is recommended. The GCPs are usually materialized by means of known points in the field. For their correct visualization in TIR images, the use of a metal plate with dimensions of at least 4 times the GSD is recommended [17].

### 2.2. Photogrammetric Processing of Thermal and RGB Images

Currently, photogrammetry is one of the most widely used techniques for the generation of 3D models [52]. Most of the commercial programs used in UAV photogrammetry are based on structure-from-motion (SfM) algorithms [53]. This technique, which is characterized by its simplicity and similarity of processing [54], is able to represent 3D information from 2D images, feature detection algorithms, and matching techniques [29]. Firstly, the software generates a sparse cloud of tie points in an arbitrary reference system, based on algorithms such as the scale invariant feature transform (SIFT) to calculate the image alignment and to determine homologous points. The point cloud is georeferenced in the next step, thanks to the EXIF metadata of the images and GCPs information. Then, the point cloud is correctly georeferenced. Multi-view-stereo (MVS) algorithms are applied to densify the cloud by using images from different points of view. Ultimately, once the point cloud is generated, the SfM software allows to obtain 3D representations, by means of meshes and textures, or 2D outputs, such as orthomosaics.

Regarding the use of UAV TIR images for the generation of thermal models, there is no standardized methodology. Throughout the literature, several alternatives can be found from authors who propose the use of TIR images only [55], mapping thermal image into 3D models [56], or the fusion of RGB and thermal images [50,57]. This research proposes a processing methodology that combines some of the aforementioned alternatives and the goodness of dual TIR sensors. The use of RGB images with two main functions is proposed. The first one is the generation of the point cloud and 3D model of the area to be represented with a higher level of detail. The second purpose is the orientation of the TIR images, so that the external orientation parameters related to the photocenter coordinates and rotation angles (yaw, pitch, and roll) are used for the alignment of the TIR images, thus solving the processing problems that usually appear when working only with this type of images [58]. Hence, it is proposed to use the TIR images to texturize the RGB model by projecting on it the temperature information, thus obtaining the thermal 3D model for the generation of the orthomosaic. The process is performed following the workflow shown in Figure 1.

### 2.3. Thermal Orthomosaic Calibration

The orthomosaic generated from the thermal images represents the value of the surface temperature for each terrain pixel. However, the accuracy of the measured surface temperature may not be as expected or as indicated by the manufacturer based on the sensitivity of the sensor. This is mainly due to the distance between the sensor and the surface to be measured, the angle of the sensor, the stability of the camera, etc. The error in the measurement can be corrected in several ways, such as calibrating the camera or measuring temperatures in the field.

Due to this, it is proposed to perform an in-situ calibration based on the measurement of temperature on different surfaces of water bodies which are represented in the orthomosaic, with another type of complementary sensor, such as thermometers or handheld cameras. The measurement of the temperature with handheld cameras of least at 3 points [59] permits to obtain higher accuracy than expected with the UAV TIR sensor [60], in addition to allowing the verification and calibration of the temperature of the water surfaces along the entire representation [61]. Complementarily, it is proposed to perform a stabilization of the thermal sensor for a few minutes before starting the flight [15,47,62].

The calibration of the orthomosaic recommends the measurement with a complementary sensor of the temperature at various georeferenced points for subsequent measurement in the orthomosaic. Hence, an average error in the measurement of the orthomosaic temperatures can be established, which will be applied by means of a reclassification with the GIS tool used for the calibration of the orthomosaic temperature.

### 2.4. Thermal Orthomosaics Analysis and Interpretation with GIS

The thermal orthomosaic represents the temperature for each point of the surveyed surface, so that, by managing the heat map generated by photogrammetric techniques in a GIS tool, a simple methodology for quasi-automatic identification of hydrothermal resources can be implemented based on the potential of GIS and photointerpretation.

Since the thermal orthomosaic represents the temperature over the entire flown surface, it is not possible to distinguish on the basis of temperature alone the water areas to be identified from other elements such as infrastructures, buildings, vegetation, etc. Due to this, the methodology proposes an analysis based on the RGB orthoimage and GIS tools. First of all, it is proposed to perform a process of photo-interpretation by visual inspection of the RGB orthoimage for the identification of the water bodies present in the surveyed site, and their extraction by means of areas or polygons with the GIS tool. From these polygons that represent the water zones, the inverse operation is proposed with respect to the thermal orthomosaic, thus extracting the water zones. This time, instead of having a raster map with information of the visible spectrum, the temperature will be obtained for each pixel. In this way, the study area for the following analyses is limited to water bodies only, avoiding possible false interpretations of other areas that may have high temperatures due to solar radiation or other factors.

At this point, the objective is the identification of potential areas that could host a geothermal resource, specifically thermal waters, so the methodology must be adapted to the regulations in force in the region in which the exploration activities are to be developed. Several approaches can be adopted for the characterization of hydrothermal waters. The first one is based on the consideration of a minimum temperature above which the effluent could be considered as such [63], while other approaches state that the temperature of the effluent must be higher than the annual average of the air temperature at its location, whereby a minimum difference may or may not be established for such categorization. Considering, for example, the case of the European Union, there is no unified criterion. Thus, in Bosnia and Herzegovina it is considered that the effluent temperature must be higher than the annual average of the air temperature in the area (without defining a value). In the cases of Austria, Czech Republic, Italy, Poland, Romania, and Slovenia it is considered that the effluent must have a temperature of 20 °C, and 30 °C in Hungary and Lithuania. Considering the regulatory framework of France, Iceland, Latvia, Portugal, and Serbia, no criterion for the definition is found [64]. In the case of Spain, the regulations are set by Law 22/1973 of 21 July, on Mines, which defines that “Thermal waters are those whose upwelling temperature is four °C higher than the annual average of the place where they emerge”. Therefore, for the identification of the potential zones, it is proposed to carry out a reclassification of the orthomosaic based on the temperature of each pixel, generating a binary map that allows to distinguish the areas whose temperatures are higher and lower than the limit that is set by the regulations. The limit temperature in Spain can be calculated with the following empirical formula:T_Limit_ = T_Avg_ + 4 °C,(1)
where T_Limit_ is the value of the limit temperature that determines the thermal character of the water, and T_Avg_ is the annual average of the place where it emerges.

The value of the limit temperature can sometimes be lower than the water temperature due to several factors such as the time of the campaign, or very cold climates. Therefore, in cases where the normal water temperature during the flight is higher, it may be advisable to apply safety coefficients and other criteria as alternatives for the determination of the limit temperature, such as the temperature of nearby hydrothermal resources, thus avoiding the identification of large bodies of water as potential hydrothermal zones.

Once the orthomosaic is reclassified and the image is obtained, it is proposed to perform a photointerpretation of the resulting potentially interesting zone and its comparison with the analog RGB orthoimage. In this way, an interpretation of the results can be made, which allows the elimination of hot zones not corresponding to geothermal resources, such as vehicles or hot areas due to solar radiation.

### 2.5. Method Validation

The proposal for validation of the methodology is based on the verification of the results in a case study. Hence, it is proposed to apply the aforementioned workflow to a particular case, on which to measure the temperature results of the possible thermal zones identified in the thermal orthomosaic, or of water bodies in case none are determined, and the subsequent field measurement of the temperatures in those areas.

## 3. Results

For the validation of the proposed method, a geological resource prospecting investigation was carried out in an area where there was already an exploited thermal water effluent, which at first glance may imply the existence of other areas of interest in its vicinity. Specifically, the case study focuses on the Deva river, as it flows through the municipality of La Hermida (Cantabria, Spain). Precisely in that place is located the Hotel-Spa La Hermida, a leisure facility intended mainly for the direct use of the geothermal resource that emanates in the area, through the implementation of spas, thermal circuits, or hot water pools. The presence of this facility allows to hypothesize the existence of other possible points of geothermal interest, as well as the opportunity to use the outdoor facilities themselves as points of measurement of water temperature for the calibration of the orthomosaic, establishing a wider range of measured temperatures.

In this location, the Deva river runs parallel to the road known as “Desfiladero de La Hermida”, which is close to the village, on the opposite bank to the aforementioned spa. Therefore, the prospection of thermal springs in the areas adjacent to the facility was proposed in the case study, by capturing thermal images with UAV about 350 m upstream and 350 m downstream of the known geothermal resource.

### 3.1. Flight Plan

The sets of RGB and TIR images were acquired with the dual sensor following a flight plan that met the aforementioned recommendations.

The UAV platform used was a DJI Matrice 600 (Figure 2), which is a drone designed for aerial photography. The platform had a WIRIS Pro dual sensor installed to capture the required set of images. The sensor characteristics are described in Table 1 and Table 2.

The flight (Figure 3) was performed according to the above-mentioned specifications, but the flight height was increased due to the presence of high voltage pylons in the study area. The dimensioning of the flight was carried out on the basis of the UgCS specific program, which allows the flight project to be developed at a constant flight height, adaptive to the variations of the terrain relief. The flight parameters are shown in Table 3.

For a correct orientation and scaling of the model, the images were georeferenced according to 18 GCPs, which were located throughout the study area, and distributed as uniformly as the accessibility of the area allowed. The GCPs consisted of photogrammetric targets represented by 30 × 30 cm steel plates with a black and white cross pattern, which are visible in both RGB and TIR images (Figure 4 and Figure 5). The coordinates of the GCPs were determined by global positioning techniques (GPS) in the official reference system (ETRS89-UTM Zone 30), using Leica GS-15 GPS equipment (Leica Geosystems, St. Gallen, Switzerland).

The flight was carried out on 10 June 2022. The acquisition of the images started at 11:00 a.m., after a 30 min period of stabilization of the thermal sensor. On the day of the flight, the weather conditions were favorable. The average temperature during the flight was 22 °C, with no wind. Due to the limitations of the batteries, the flight was carried out in four independent and subsequent flights. In addition, following the recommendations for the thermal sensor, its emissivity was set to 0.98. The WIRIS Pro sensor directly stored the surface temperature value for each pixel, generating the TIR images, which were stored in TIFF format.

### 3.2. RGB and TIR Photogrammetry

The set of images was processed according to the workflow proposed above. The management began with the processing of the set of 1164 RGB images in JPEG format, following the conventional methodology, which includes the orientation of the images, georeferencing and generation of the point cloud, the meshes, and the orthomosaic (Figure 6).

Once the management of the RGB images was completed, the 1164 thermal images in TIFF format were processed. The first step was to import the information related to the external orientation of the RGB images, which was assigned to their homologous thermal images in order to solve the orientation problems that appear when working with this type of images. Next, georeferencing was carried out with the GCPs, and the point cloud and 3D model were generated. The process ended with the generation of the thermal orthomosaic (Figure 7), which due to the image format directly represents the surface temperature value for each pixel.

The processing of the photogrammetric block of images was developed with Agisoft Metashape, version 1.7.2 (Agisoft, LLC. St. Petersburg, Russia).

### 3.3. Thermal Orthomosaic Calibration

Due to the great distance between the thermal sensor mounted on the UAV and the surface of the water bodies, deviations in the temperature measurement are generated. To solve this problem and obtain a representation more in line with reality, the temperature was taken at nine points along the study area, which were used to evaluate and correct the deviation between the UAV measurement and that of the complementary sensors used.

In the field, two complementary types of sensors were used for in-situ temperature measurements: mercury thermometers and a FLIR E40 handheld camera. The measurements with both sensors provided very similar temperatures for the evaluated points and considering the lower precision of the mercury thermometer due to the division of its scale in units of centigrade degrees, the handheld camera measurements were used as reference. In addition, this sensor allows to measure temperature in a greater number of points, where the difficult access to the river complicates the use of the thermometer. The temperature measurement points for calibration (Table 4) were geo-referenced with GPS where possible (Figure 8). In areas of difficult access, they were measured only with the handheld sensor. The area closest to the riverbank was georeferenced for later identification in the thermal orthomosaic.

The average difference between the UAV and the handheld sensor measurements is 3.506 °C, with a deviation of 0.955 °C. The UAV always provides a lower temperature. Therefore, the calibration of the orthomosaic is performed by increasing the temperature of the pixels by 3.506 °C throughout the entire representation. This operation can be performed by a raster transformation in Agisoft at the end of the photogrammetric process, or by a reclassification of the entire image using the GIS tool. In this case, the second option was used, employing the open-source program QGIS for the calibration of the thermal image and the subsequent analysis for the identification of the potential zones of interest. The calibrated orthomosaic obtained is shown in Figure 9.

### 3.4. GIS Analysis

Once the corresponding RGB and thermal orthomosaics were obtained, they were analyzed by means of QGIS, which is the tool selected for the identification of zones with potential hydrothermal upwelling.

First of all, a photointerpretation of the orthoimage was developed for the segmentation of the water bodies. Hence, the quasi-automatic analysis could be performed only on the water surfaces, eliminating possible errors due to the temperature of other elements present in the area such as the road, vehicles, buildings, etc. The segmentation was carried out through a visual analysis and the generation of a layer of polygons that delimited the water bodies present, the riverbed, and the pools of the spa facilities (Figure 10).

On the thermal orthomosaic, an extraction was performed based on the previously generated polygon layer, so that the temperature distribution along the entire surface of the represented water bodies was obtained (Figure 11).

The last operation to be carried out was a reclassification of the surfaces according to the previously defined limit temperature. Based on the regulations of the Mining Law in force in Spain, it was determined that the limit temperature was 16.9 °C. The average annual temperature data for the area were provided by the State Meteorology Agency (AEMET), based on the historical records of the Climate Atlas, which allow obtaining that information for a selected area or polygon. Since, according to current legislation, the average temperature of the river itself on the day of the flight was higher, the average temperature of the spa pools, which pump water from wells, that is to say, directly from the resource itself, was adopted as limit temperature. Hence, the limit temperature considered was 32.75 °C. This variation makes it possible to focus quasi-automatic identification in areas where the temperature is higher, and therefore, of greater potential.

The reclassification made it possible to generate a binary map, in which the potential geothermal interest of every pixel was evaluated according to their temperature value. The pixels with a temperature higher than the limit value were distinguished in one color, and the zones of no interest, where the temperature is lower than the limit, were represented in another color (Figure 12).

Once the binary map was obtained, all the potential zones were visually analyzed according to the chromatic differences established by the reclassification symbology. Figure 12 shows how the reclassification was carried out by setting in blue the zones whose temperature was lower than the limit, and in green those of higher temperature, which are therefore the zones of potential interest. This palette is customizable and can be modified on the basis of the most contrasting colors to facilitate the visual identification of the potential areas.

In order to conclude, the proposed methodology required a visual inspection of the potential geothermal zones. A simple comparison between the reclassified image and the orthomosaic should be carried out by an overlay of raster layers georeferenced in QGIS. This final step allowed to remove false geothermal interest areas without a field inspection, as elements present on the water surface, such as rocks or logs, which may have been previously classified as water.

Figure 13 shows the presence of a large number of green spots. However, most of them correspond to rocks, or the area where the riverbed comes to the surface, in the middle of the channel.

### 3.5. Validation of Results

After analyzing the orthoimages with QGIS, it was possible to identify an area of potential hydrothermal interest located in the central zone of the surveyed area, specifically under the bridge that provides access to the spa facilities. The zone of interest can be seen in Figure 14. According to the information obtained from the orthomosaic, the temperature in the zone of interest is 39.937 °C, which is 7.187 °C above the set limit temperature.

In order to validate the result, another trip was made to the field to measure the temperature at that point. The results of this field visit made it possible to determine the temperature of the zone with potential interest, obtaining a temperature of 40 °C with the thermometer, and 40.9 °C with the handheld camera (Figure 15 and Figure 16). The final thermal orthomosaic shows an accuracy of about 1 °C, and a deviation of 1.037 °C with respect to the measurements obtained with the complementary sensors used to verify the presence of the geothermal resource. This measurement allows validating the temperature obtained from the thermal orthoimage and verifying the existence of a hydrothermal effluent.

## 4. Discussion

After the application of the proposed methodology for the identification of zones with potential hydrothermal interest, and in view of the results obtained, the following interpretations can be presented:Flight planning is fundamental, not only in terms of flight parameters, but also with respect to the date of the activities. On the one hand, the possible vegetation present in the area must be considered. Therefore, flights in autumn, when the trees are leafless, can favor the capture of images by eliminating the foliage, as well as a better photointerpretation of the result. In addition, the flow of the river must be considered, as an excessive amount of water can camouflage the thermal effluent, and its absence could lead to its interpretation as a free-of-water-surface area. Both circumstances would not allow its identification, which implies that a medium flow may be the best alternative.The design of the flight plan must be done in a more thorough way than in the case of RGB-only flights. This observation must be taken into account in order to avoid later processing problems due to the lower resolution and contrast of the TIR images. In addition, the correct configuration of the sensor must be considered, establishing the appropriate emissivity or ambient temperature parameters, as well as a correct storage format of the TIR images data, in order to obtain the surface temperature information that allows getting an adequate result.The workflow based on the complementary processing of both blocks of analog images allows to obtain a high-quality representation of the study area, despite the lower resolution of the thermal sensor, which is only used for the projection of the temperature distribution information.The measurements of the thermal sensor of the dual device show a large average difference of 3.506 °C with respect to the measurements of the rest of the equipment applied. In spite of establishing the same configuration as in the handheld camera and having a stabilization period prior to the start of the flight, the UAV always measured lower temperatures. This condition implies the need to use temperature calibration points during the flight in order to reduce the error in the final result.The photointerpretation of images for the first identification of water sheet surfaces is a process that can be time consuming. Therefore, the implementation of automatic or quasi-automatic recognition techniques can speed up and simplify the overall process.GIS tools allow various operations, such as the analysis and management of photogrammetric products obtained for the identification of hydrothermal resources. The use of this type of photograms facilitates the photointerpretation and visualization of the data obtained, allowing the evaluation of the result. It is a tool with great potential that can provide a greater simplicity to the proposed methodology if the ways to automate some of the tasks performed are investigated.The proposed methodology does not involve actions of great complexity. Although it simplifies the process with respect to traditional techniques, providing a solution with lower economic and time costs, it requires a second field visit to evaluate and verify the results obtained.Considering the application of the methodology to the case study, it allows the identification of a thermal anomaly in the water bodies present in the area. This methodology measures the water temperature at an isolated moment when the photogrammetric flight is performed. Therefore, it would be necessary to carry out field trips throughout the year to assess the conformity of the temperatures to the specifications set by current legislation.

The results obtained have made it possible to identify a zone of potential interest in the vicinity of the spa, with a water temperature of over 40 °C. In spite of being a low temperature upwelling, these indications may allow the subsequent development of more expensive prospecting techniques, such as boreholes, to evaluate the evolution in depth of the potential geothermal interest of the resource in that area, and its possible exploitation.

## 5. Conclusions

The potential of geothermal resources represents a great opportunity for the valorization of the resource depending on the temperature of the specific upwelling. Conventional prospecting techniques for this type of resources are costly and time consuming, so the proposed methodology, which is based on the use of UAV platforms and dual sensors, is a feasible and reliable alternative for this type of activities.

This methodology is a set of simple exploration procedures that allow a rapid quasi-automatic identification of potential hydrothermal upwellings. The use of photogrammetry with images captured with UAV allows the generation of thermal maps which can be easily interpreted thanks to GIS tools, favoring the identification of thermal anomalies with exploitation potential. In this way, subsequent prospecting activities can focus on evaluating the real capacity of the resource.

The versatility of the proposed methodology should be highlighted, due to the possibility of adapting the limit temperature for reclassification according to the particular conditions of the area, or the legislation in force in the country of application. This versatility grants the opportunity to perform different tests by modifying the limit temperature immediately. This fact offers great flexibility to the methodology and allows different interpretations on the same zone based on the criteria to be set. In this way, the methodology can be extrapolated not only for the prospection of geothermal or hydrothermal resources, but also for any other area where the knowledge of water temperature implies a positive impact, or for the management of multi-criteria analysis with GIS tools.

The proposed methodology could be implemented for other applications or lead to future research lines. Hence, following the same methodology, it could be possible to monitor the hydrothermal resources by capturing periodic images that allow to represent and assess the state of the effluent, or to implement algorithms for the automatic identification of water surfaces to speed up the process.

In short, the alternative of implementing UAV platforms with dual sensors, photogrammetric techniques, and GIS tools is a faster, cheaper, and simpler alternative than conventional techniques. It allows obtaining high quality results and can be adapted to cases located in complex or difficult to access areas.

## Figures and Tables

**Figure 1 sensors-22-06756-f001:**
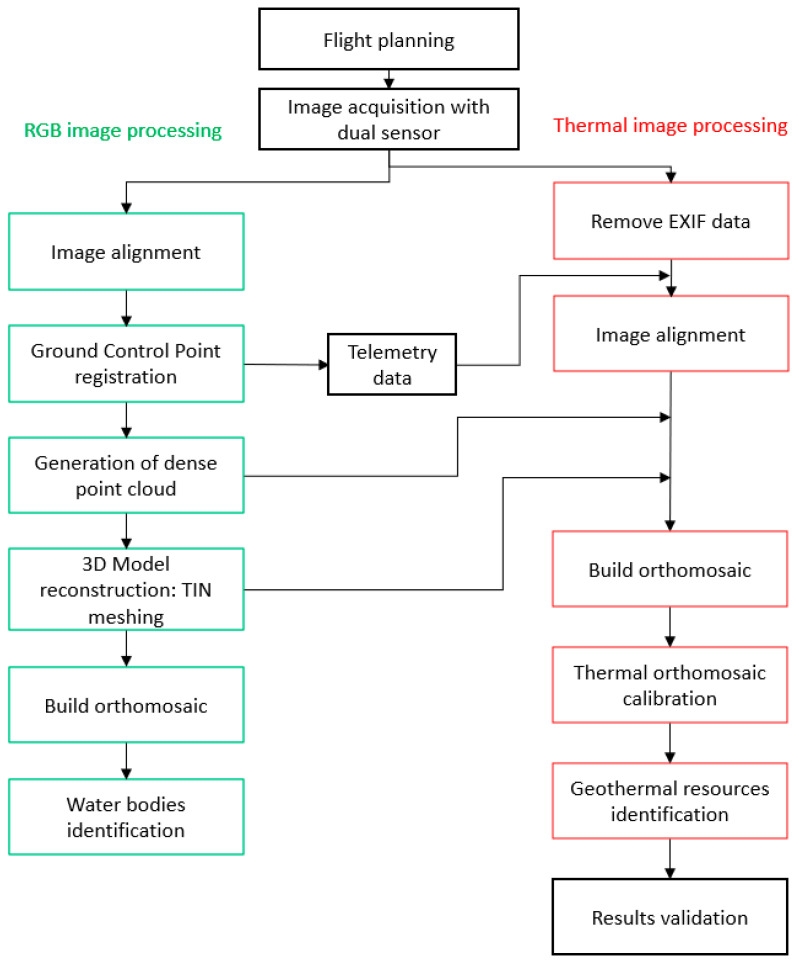
Proposed workflow.

**Figure 2 sensors-22-06756-f002:**
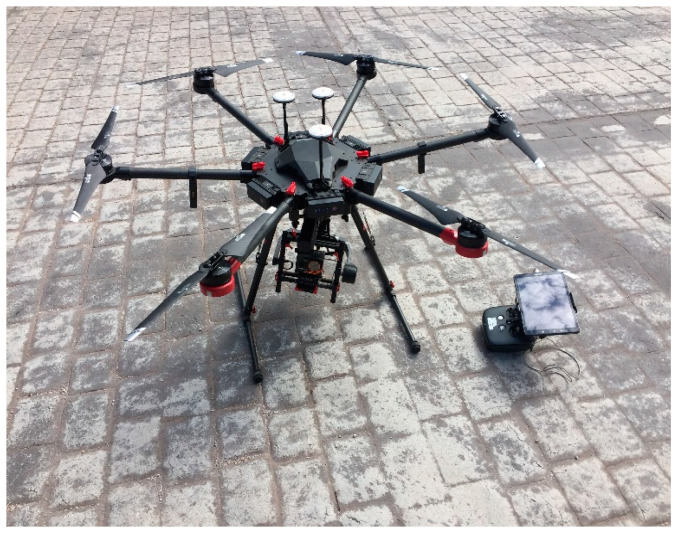
UAV DJI Matrice 600 and WIRIS Pro dual sensor used for image capture.

**Figure 3 sensors-22-06756-f003:**
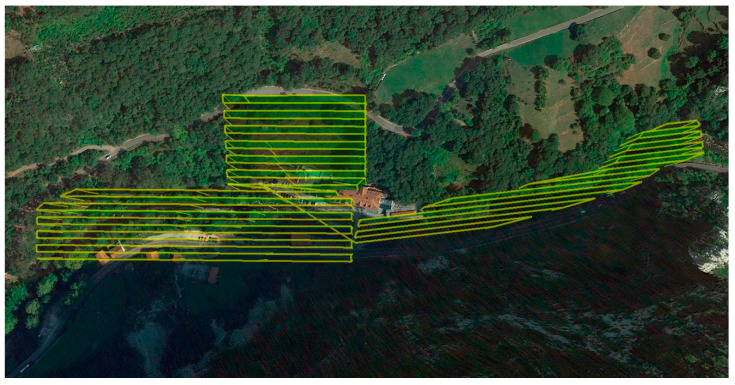
Flight geometry on simple grid configuration performed over the riverbed and spa facilities.

**Figure 4 sensors-22-06756-f004:**
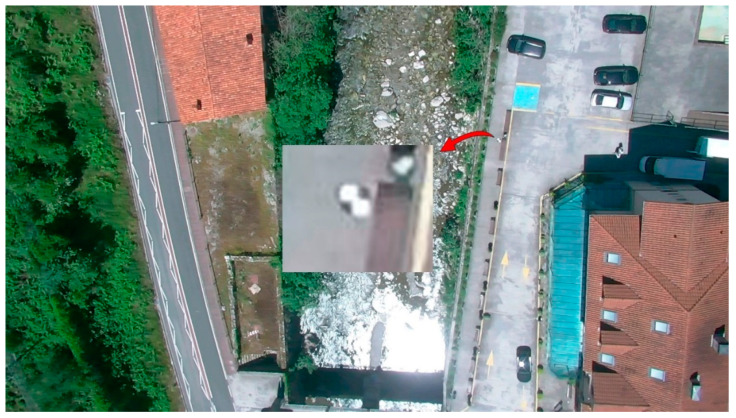
Detail of a GCP on RGB image.

**Figure 5 sensors-22-06756-f005:**
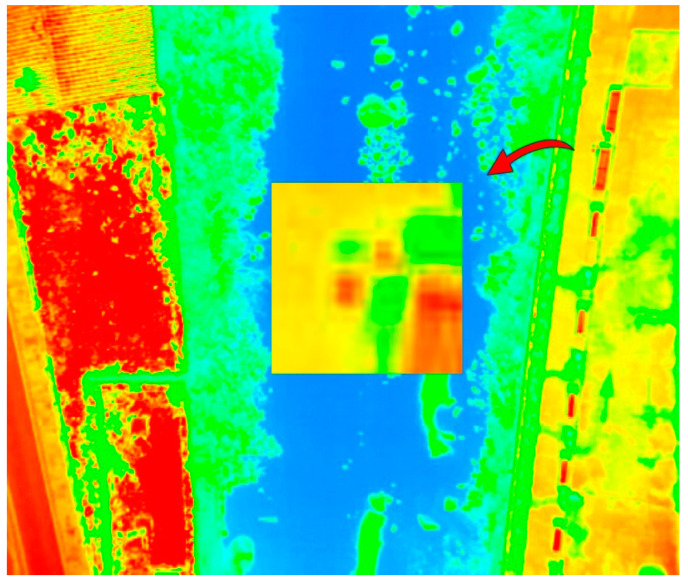
Detail of a GCP on thermal image.

**Figure 6 sensors-22-06756-f006:**
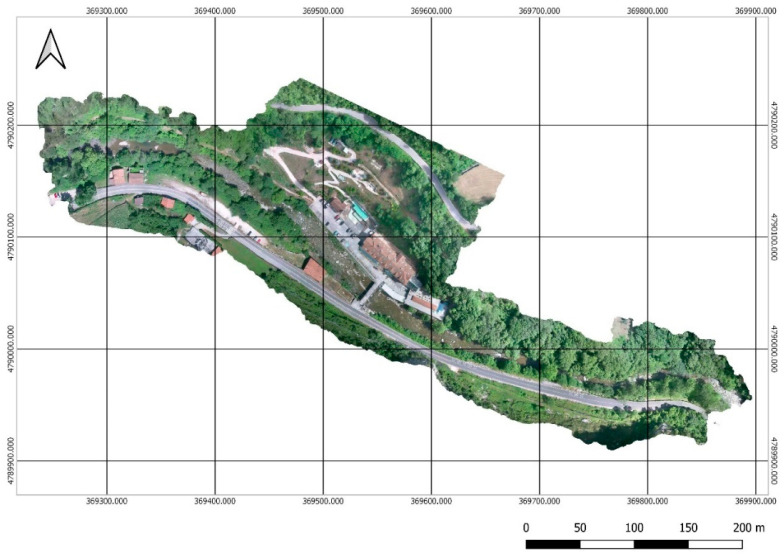
RGB Orthomosaic with a GSD of 4.27 cm/px.

**Figure 7 sensors-22-06756-f007:**
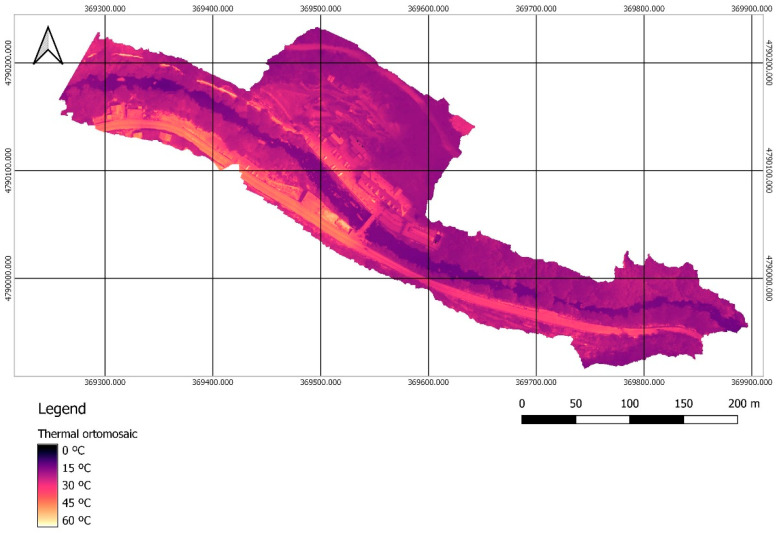
Thermal orthomosaic for the surface temperature, with a GSD of 6.26 cm/px.

**Figure 8 sensors-22-06756-f008:**
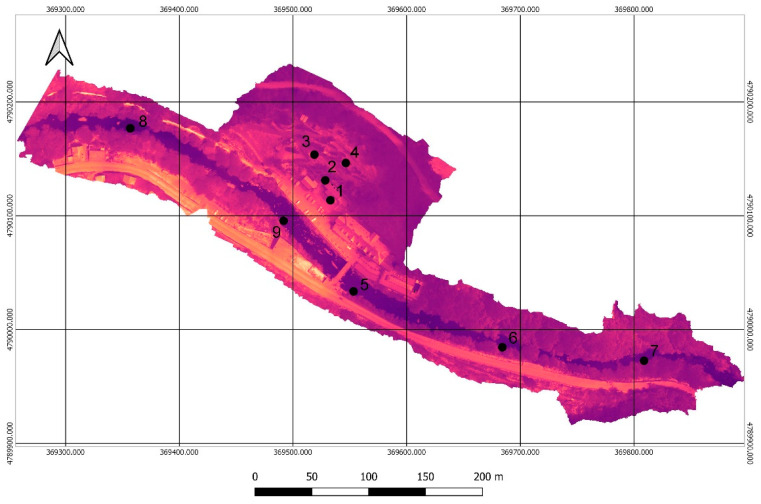
Georeferenced points for thermal calibration of the thermal orthomosaic.

**Figure 9 sensors-22-06756-f009:**
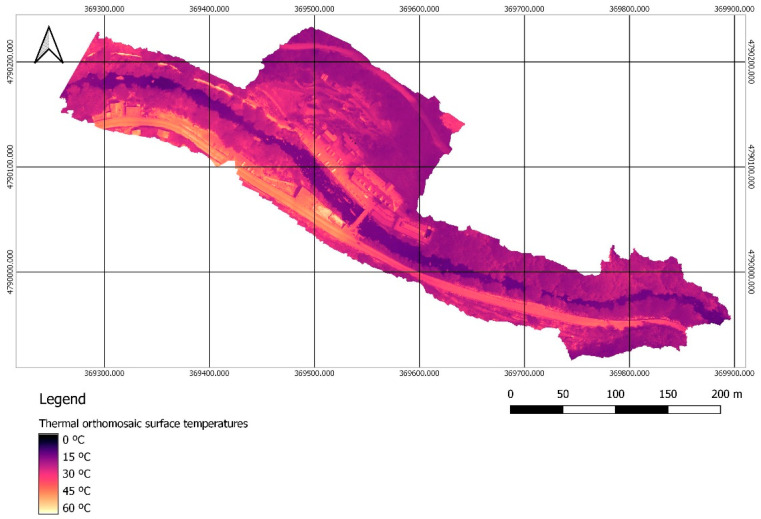
Thermal orthomosaic calibrated with the handheld sensor in-situ measurements.

**Figure 10 sensors-22-06756-f010:**
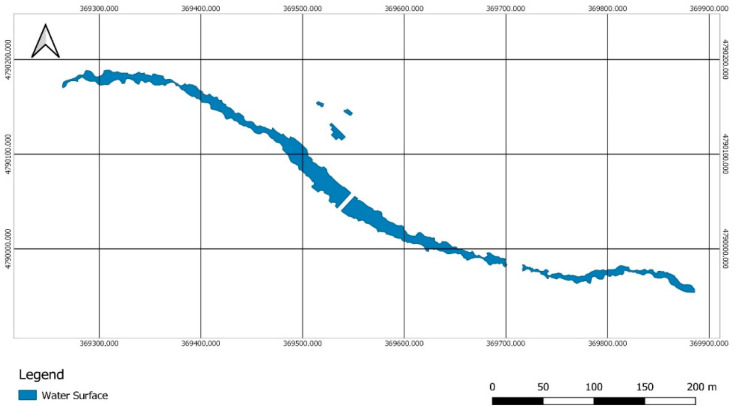
Water surface polygons. Segmentation of the water surfaces of the river and spa facilities based on a visual analysis of the RGB orthoimage.

**Figure 11 sensors-22-06756-f011:**
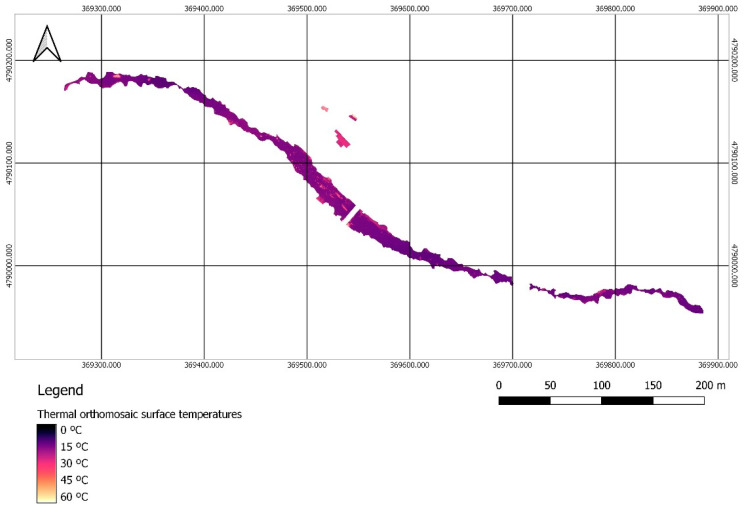
Thermal orthomosaic of the water surface. Determination of the surface temperature of the areas of interest by extraction of the water surface polygons from the thermal orthoimage with GIS.

**Figure 12 sensors-22-06756-f012:**
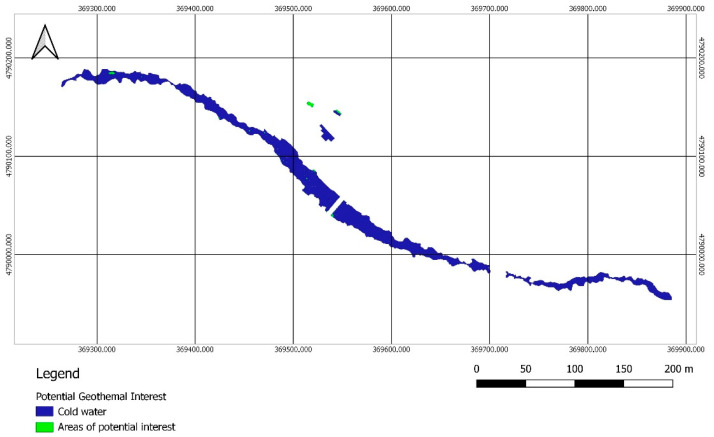
Orthomosaic representation of the potential geothermal interest areas. Reclassification for the set limit temperature of 32.75 °C. The areas of potencial interest are shown in green, while the cold water surfaces are represented in blue.

**Figure 13 sensors-22-06756-f013:**
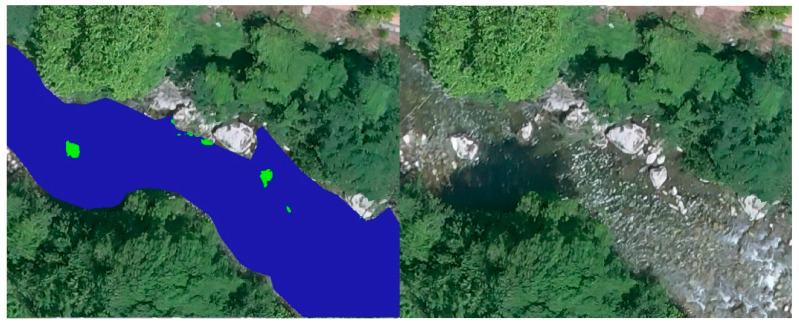
Error zones.

**Figure 14 sensors-22-06756-f014:**
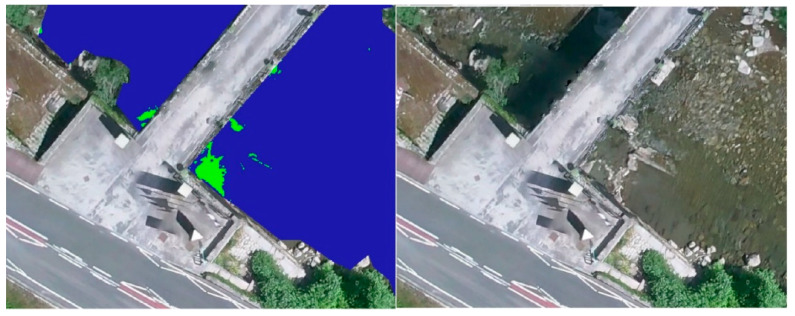
Reclassification detail and RGB orthomosaic zoom of the area of potential interest. The image allows assessing that an area of potential interest corresponds to the surface of a water body, although it is still necessary to check the temperature in the field.

**Figure 15 sensors-22-06756-f015:**
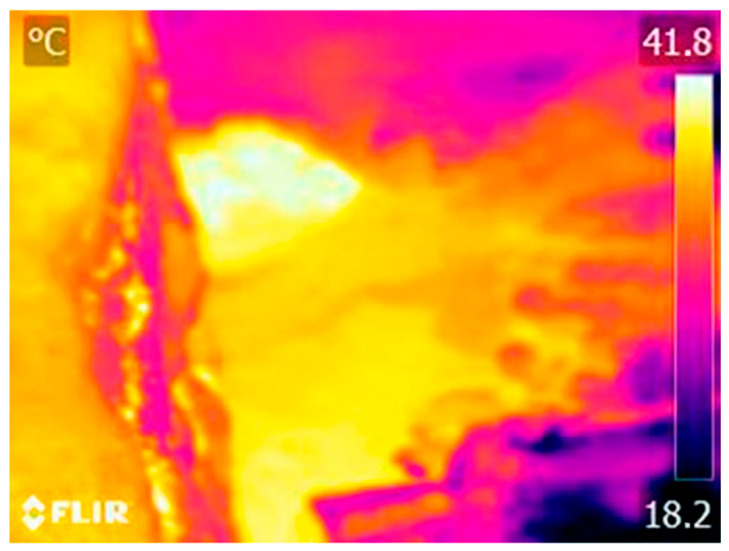
Thermal image of the area with potential interest taken with a FLIR E40 handheld sensor to validate the final results.

**Figure 16 sensors-22-06756-f016:**
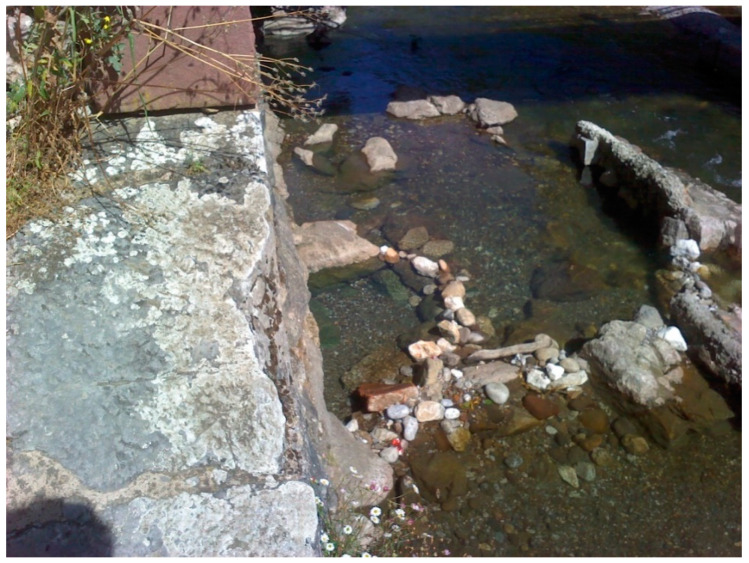
RGB image of the area with potential interest taken with a FLIR E40 handheld sensor to validate the final results.

**Table 1 sensors-22-06756-t001:** RGB dual sensor specifications.

Specifications	
Sensor size	1/3”
Resolution	1920 × 1080 px
Focal distance	4 mm
Format	JPEG

**Table 2 sensors-22-06756-t002:** Thermal dual sensor specifications.

Specifications	
Sensor type	Uncooled Vox microbolometer
Sensor size	1/3”
Resolution	640 × 512 px
Focal distance	19 mm
Spectral range	7.5–13.5 μm
Temperature sensitivity	0.05 °C
Accuracy	±2 °C
Format	JPEG

**Table 3 sensors-22-06756-t003:** Flight plan parameters.

Flight Parameters	
Height (m)	70
Speed (m/s)	3
Frontlap (%)	80
Sidelap (%)	80
Resolution (cm/px)	2.23 (RGB) 6.12 (IR)
Number of Photographs	1164
Flight time (min)	96

**Table 4 sensors-22-06756-t004:** Measurements for the thermal calibration.

Point	Thermometer (°C)	Handheld Camera (°C)	UAV (°C)
1	26	26.5	23.95
2	28	28.8	25.70
3	35	35.3	31.25
4	42	40.4	38.43
5	18	17.1	14.10
6	-	17.8	12.63
7	-	16.4	13.32
8	-	17.7	13.95
9	19	18.2	14.33

## Data Availability

The data presented in this study are openly available in the repository of the University of Cantabria, at: http://hdl.handle.net/10902/25534 (accessed on 3 September 2022).

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
