# Peer review of "Thermal Water Prospection with UAV, Low-Cost Sensors and GIS. Application to the Case of La Hermida"

_sensors, 2022, doi:10.3390/s22186756_

Round 1

Reviewer 1 Report

The paper presents good idea ,but  the paper needs  some comments for publication as following:

1.Adapt and  rewrite the abstract

2.Enhancement the grammar of the paper

3.Make a lot of experiments

4. No comparison with prior arts.

5.Explain with details  the experimental results and analysis

6.Rewriting the Conclusion and add the future work.

7.Adapt the references

Reviewer 3 Report

The authors describe a case study (in La Hermida, Cantabria, Spain) in which geothermal resources are analysed by means of a field survey using an infrared sensor mounted on a UAV.   The article is well written and therefore I suggest publication after a few revisions.   - A summary of the regulations (not only in Spain) in the introduction could be useful to understand the applicability of the data processing presented also in other cases.   - The measurements for thermal calibration (Table 4) must be reported on the map. In addition, the deviation from the average could be indicated in the text.   - The map legends (figures 7,8,10) are to be reviewed, I guess the classes represent the temperature range and not single values. However, the limits in Figure 10 need to be changed for greater readability.   - Figure 14 is unclear, perhaps the authors should add the corresponding RGB image.   - Authors should discuss the reliability due to a single survey period, probably surveys in different periods or the comparison between day and night could provide more information.

Round 2

Reviewer 4 Report

Accept the paper